# Residual Stress Redistribution Analysis in the Repair Welding of AA6082-T6 Aluminum Alloy Joints: Experiment and Simulation

**DOI:** 10.3390/ma15186399

**Published:** 2022-09-15

**Authors:** Zhihao Chen, Yanjuan Duan, Ping Wang, Hongliang Qian

**Affiliations:** 1School of Civil Engineering, Harbin Institute of Technology, Harbin 150001, China; 2The 41st Institute of the Fourth Academy of CASC, Xi’an 710000, China; 3School of Ocean Engineering, Harbin Institute of Technology at Weihai, Weihai 264209, China

**Keywords:** three-dimensional scale effect, repair welding, residual stress redistribution, optimal repair principles, BS7910

## Abstract

Residual stress has a three-dimensional scale effect (length, depth, and width) in the process of repair welding, which has a detrimental impact on the service of the aluminum alloy welded structures in high-speed trains. This paper aims to systematically analyze the effects of the repair welding dimension on the residual stress redistribution and obtain the optimal repair welding principles. A combination of blind-hole drilling method and stress linearization in BS7910 was adopted to investigate residual stress redistribution under various repair welding dimensions. The results indicate that repair welding dimension was in accordance with the principle of “SNL (shallow, narrow and long)” and the optimal repair length, depth, and width of butt joints in this study were 15t, 0.25t, and t, respectively (t is the plate thickness of butt joints).

## 1. Introduction

AA6082-T6 aluminum alloy is a crucial part of the lightweight design used in high-speed train bodies [1]. Since high-speed strains are in operation at a speed of more than 300 km/h, aluminum alloy welded structures endure various complex loads. Meanwhile, with the increase in the service time, various defects in the welded structure gradually begin to be exposed. Therefore, scientific repair welding of structural defects in the aluminum alloy car body need to be solved urgently.

Repair welding not only prolongs the service life of welded structures but also saves costs, which has been widely used in many aspects of actual production. However, repair welding is a process of localized reheating and cooling of welded structures, which affects the microstructure of the repair welding area [2,3,4]. Li [5] found that crystals refinement and a preferred orientation appeared in the nugget zone when friction plug repair welding was conducted. Meanwhile, repair welding will reduce the mechanical properties of the joint to varying degrees, such as: fatigue properties [6], microhardness [7,8,9], fracture toughness [10], and susceptibility to stress corrosion cracking [11]. Kang [12] found that stress rupture life sharply reduced after repair welding, local residual stress and crystal orientation would lead to the repair welding cracking. Zhang [13] investigated the mechanical property of the friction stir lap repair welding and found that the hardness in the nugget zone and fracture strength was significantly improved. Marenych [14] discovered that the hardness of both deposition and component increased after annealing + ageing by 65% than that of the as-welded condition when wire arc additive manufacturing was conducted to repair welding.

In addition, repair welding causes redistribution of the as-welded welding residual stress, and localized repair welding has a more serious impact on welded structures, especially in the weld metal and heat affected zone (HAZ) [15], which can be measured and analyzed based on multiple-cut contour method [16], neutron diffraction [17,18], x-ray diffraction method [19], blind-hole drilling method [20], and the finite element method (FEM) [21]; among these methods, blind-hole drilling method and the FEM are the most common methods. The residual stress causes weld zone of structures to enter the yield stage prematurely with the superposition of external load. How to reduce the repair residual stress, researchers have carried out research from the aspects of repair process [22], repair method [23,24], repair sequence [25], environment, and treatment before and after repair welding [26,27,28]. Zhang [29] found that the residual stress generated by the high-energy spark deposition method for weld repair was low and only concentrated in the repair weld. Chu [30] applied compressive stress to the defect location of the pressure piping using overlay welding. Hasheemzadeh [31] considered the steel aging and corrosion environment in the repair process. Charkhi [32] found that increasing the preheating temperature of repaired welded pipe reduced the longitudinal residual stress on the inner and outer surfaces of the steel pipe by approximately 35–50%, respectively. Aung [33] discovered that post-defect repair heat treatment (PWHT) would release the residual stress and improve fatigue performance of repaired joint.

Although considerable research has been devoted to studying the repair welding length [34] (or depth [35], or width [36]) from different perspectives for different joint forms, rather less attention has been paid to systematically analyzing the effect of repair welding dimension on residual stress redistribution. In this paper, the combination of experimental tests and FEA was used to fully validate the accuracy of FE models of butt joints and the heat source models, and based on the principle of stress linearization in BS7910, the residual stress variation at the weld centerline (WCL), weld toe (WT), and longitudinal centerline on upper surface for butt plates before and after repair welding was systematically analyzed; finally, the optimal repair welding principle was given.

## 2. Materials and Methods

### 2.1. Preparation of Test Specimens

In this study, AA6082-T6 and ER5356 aluminum alloy were used as base metal and welding wire, respectively, and their chemical compositions and mechanical properties are shown in Table 1 and Table 2.

Melt inert-gas (MIG) welding equipment was used for aluminum alloy butt welding with the advantages of fast welding speed, low energy input, and high efficiency. As shown in Figure 1, as-welded butt plates (350 mm × 300 mm × 8 mm (length × width × thickness)) were obtained by multi-layer multi-pass welding of two aluminum alloy plates with the same size (350 mm × 150 mm × 8 mm). In the repair welding process, the weld material with the approximate size of 200 mm × 8 mm × 4 mm (length × width × depth) in Figure 1b was first excavated and then the repair welding specimens in Figure 1c were obtained by manual welding.

### 2.2. Stress Linearization According to BS7910

The effect of residual stress with high nonlinearity on welded structural integrity cannot be characterized intuitively and accurately. Therefore, we can conduct research by means of the treatment measure of secondary stress *Q* in integrity assessment, which is also an important parameter to evaluate the repair welding effect. As shown in Figure 2, according to the linear decomposition principle of residual stress in the BS7910 standard [37,38], the residual stress distribution along the crack propagation path is decomposed into membrane stress *σ_m_*, bending stress *σ_b_* and self-equilibrium stress *σ_sb_* based on Equations (1)–(4).
(1)$$\sigma_{R}=\sigma_{Y}\left[a+a_{1}\left(\frac{z}{B}\right)+a_{2}{\left(\frac{z}{B}\right)}^{2}+a_{3}{\left(\frac{z}{B}\right)}^{3}+a_{4}{\left(\frac{z}{B}\right)}^{4}\right]$$
(2)$$\sigma_{m}=a_{5}\sigma_{Y}$$
(3)$$\sigma_{b}=\sigma_{Y}\left[a_{6}+a_{7}\left(\frac{z}{B}\right)\right]$$
(4)$$\sigma_{sb}=\sigma_{R}-\sigma_{m}-\sigma_{b}$$
where *σ_R_*, *σ_m_*, *σ_b_*, *σ_sb_*, *σ_Y_* are residual stress, membrane stress, bending stress, self-balancing stress component, and yield strength, respectively; *a*, *a*_1_, *a*_2_, *a*_3_, *a*_4_, *a*_5_, *a*_6_, *a*_7_ are the fitting parameters of residual stress, membrane stress, and bending stress, respectively, the value of which can refer to BS7910. *B* and z are section thickness in plane of flaw and measure of position through the thickness, respectively.

Membrane stress *σ_m_* and bending stress *σ_b_* are the main parameters that lead to fatigue failure of welded components. According to the BS7910 standard, it is necessary to consider the variation in membrane stress and bending stress caused by repair welding, when performing defect safety evaluation.

## 3. Results and Discussion

### 3.1. Repair Welding Schemes

For the research on repair welding schemes of butt joints, the residual stress distribution under different repair lengths, depths, and widths was considered, with a particular emphasis on the variation in membrane and bending stress. The specific repair schemes are as follows:(1)Repair welding length schemes

Three group schemes were designed with the parameter of plate thickness *t* when the length repair was studied (the plate thickness *t* of butt joints is 8 mm). The repair width was set as t, the repair depth was t/2, and the repair lengths were 4t, 10t, and 15t, respectively (see Table 3).

(2)Repair welding depth schemes

Three group schemes were also determined when studying the repair depth (see Table 4); the repair length was set as 15t, the repair width was t, and the repair depth were 0.25 t, 0.5t, and 0.75t, respectively.

(3)Repair welding width schemes

As shown in Table 5, on the premise that the repair length and depth were 15t and t/2, respectively, three group schemes with a repair width of t, 1.5t, and 2t were designed.

According to the above repair welding schemes, the analysis methods were formulated as follows:(1)Analysis of the longitudinal residual stress (LRS) and transverse residual stress (LRS) nephograms under different repair length, depth and width, as shown in Figure 3;(2)Analysis of the temperature variation at different positions of butt joints during the welding process;Analyze the temperature variation of three points at the longitudinal centerline on the upper surface of butt joints with time, as shown in Figure 3 (Point1: at WCL; Point2: at WT; Point3: at HAZ), and validate the accuracy of heat source models in FEA;(3)Analysis of longitudinal and transverse residual stress distribution at WCL and WT.According to the principle of stress linearization, the LRS and TRS distribution at WCL and WT under different repair lengths, depths, and widths was extracted and analyzed;(4)Analysis of membrane and bending stress distribution in the longitudinal and transverse directions at WCL and WT;(5)Analysis of the LRS and TRS distribution at the longitudinal centerline on the upper surface of butt plates (i.e., path 1);(6)According to the comprehensive analysis of simulation results, the optimal repair length, depth, and width were determined.

### 3.2. Finite Element Modeling for Repair Welding

In this study, the finite element (FE) software ABAQUS was used to investigate the effects of repair length, depth and width on the residual stress distribution. The specific simulation procedure is as follows:(1)As-welded FE modeling

The dimension of butt joints is 300 × 200 × 8 (length × width × thickness), which was welded by two layers and three passes welding. The instantaneous heat source models were adopted during the thermal-mechanical sequential coupling process, DC3D8 and C3D8 elements were used in thermal analysis and force analysis, respectively. Three dimensional (3D) solid models of butt joints are shown in Figure 4. In order to simulate the welding process more accurately and to improve calculation efficiency, non-uniform meshes were arranged; the mesh size of the weld seam and base metal at the distal end was set as 0.6 mm and 4 mm, respectively. there was a total of 297,955 meshes and 327,874 nodes in the FE models. The physical properties of materials used in this study can refer to the work of Yu [40]. The boundary conditions of FE models are shown in Figure 4c. The movement of four nodes at the left end and the transverse centerline at the bottom of the FE model was restricted in the Z and X directions, respectively, and the freedom degree in the Y direction of two columns nodes symmetrical to the transverse centerline at the bottom of the FE model was constrained.

(2)Analysis steps

The analysis steps were set in ABAQUS, and in order to ensure the accuracy of FEA, the specific welding parameters of butt joints in the heating and cooling process were determined in Table 6, the instantaneous heat source model was selected in this study. During the welding process, each weld bead was heated to the melting temperature of the aluminum alloy (660 °C) within 3 s.

(3)Repair welding modeling

In the process of repair welding, the mesh size and number of the repair welding model and the as-welded model were consistent, the repair welding area was determined by redefining the birth-death elements, so that stress and strain distribution of as-welded butt joints can be imported into repair welding models using the keyword editing function in ABAQUS with the following codes:

* Initial Conditions, type = Stress, Input = Residual Stress.csv

* Initial Conditions, type = Hardening, Input = PEEQ.csv

Among them, “Residual stress” and “PEEQ” are user-defined file names, and the data in the csv file includes S11, S22, S33, S12, S13, S23 and PEEQ.

The geometry of the parts, physical properties and boundary conditions in the finite element simulation of repair welding were the same as that of as-welded butt joints, and the instantaneous heat source was also adopted in repair welding models. In order to ensure the consistency of the input heat for each repair welding model, when studying the effect of repair welding depths D1 (0.25t), D2 (0.5t), and D3 (0.75t) on the residual stress, single-pass welding, two-pass welding, and three-pass welding were used, respectively, the analysis procedure of the repair welding width was consistent with that of repair welding depth.

In this study, the residual stress of the original welding state was firstly imported and then the repair weld beads were killed. It can be seen from Figure 5 that residual stress of FE models after killing weld beads was released to a certain extent, which is consistent with the actual situation.

Blind-hole drilling tests were used to validate the accuracy of FEA, as shown in Figure 1b, strain gauges arranged near the weld metal were relatively dense when the residual stress was measured due to drastic change of residual stress. Comparing the results of the LRS and TRS distribution in FE models after repair welding with experimental tests (see Figure 6), it can be seen from Figure 6 that the peak value of the LRS after repair welding reached 225 MPa, the variation trend of residual stress obtained by the blind hole-drilling tests and the simulation was consistent, and the value at high residual stress area was very close; the difference in values was mainly reflected in the low stress area due to the fluctuations in data, the FEA data was in good agreement with the test data. Therefore, the accuracy of repair welding models was validated.

### 3.3. Effects of Repair Welding Length

According to the LRS distribution of FE models under different repair lengths in Figure 7, the entire weld was in a high longitudinal tensile stress state, and with the increase of repair welding length, the yield area after repair welding gradually expanded. Meanwhile, the high-pressure stress area on the left and right sides of butt plates also gradually became greater. The residual tensile stress in the weld area was very significant with the value of approximately 300 MPa, which then in the HAZ decreased rapidly to 50–100 MPa with the increase of the distance to the weld area; immediately afterwards, the LRS gradually changed from tensile stress to compressive stress, and finally the compressive stress at both ends reached approximately 200 MPa.

From Figure 8, as repair welding length grew, high tensile stress area at the arc starting and ending of the repaired weld gradually declined, and the compressive stress at both ends of original weld gradually went down with the appearance of compressive stress. When the repair welding length was 4 t, the entire repair weld area of butt joints exhibited high concentrated transverse tensile stress.

To observe the temperature variation at three points in Figure 2 more intuitively, only the first 15 s were taken for each welding due to the drastic changes in temperature (see Figure 9). It can be seen that the temperature variation corresponding to the three points were basically the same. First, the temperature at WCL was the highest, followed by the WT, and the temperature at HAZ was the lowest, which was consistent with the actual welding situation.

The LRS and TRS distribution at WCL and WT in the FE models under different repair lengths are shown in Figure 10 and Figure 11, respectively. The residual stress at WCL first slowly increased, then steeply fell and after that rose. On the whole, the residual stress with the repair length 4t was the greatest, followed by the repair length of 10t, and the repaired length of 15t was the minimum.

The membrane and bending stress at WCL and WT were linearized to obtain Figure 12; with the increasement of the repair length, the transverse membrane stress was greatly reduced, and transverse membrane stress for repair welding length L3 (15t) was only 23.4% of that for repair welding length L1 (4t). The transverse bending stress maintained at a high value. However, the longitudinal membrane and bending stress were both small. Since the transverse membrane stress would accelerate the failure of welded structure, the repair length should be as long as possible.

As can be seen from Figure 13a, repair welding length had little effect on the LRS distribution. However, the TRS at the weld zone and HAZ fell off with the growth of repair welding length (see Figure 13b).

It can be basically seen from the simulation results that the length corresponding to the best repair effect was 15t among the repair welding lengths of 4t, 10t, and 15t. In the process of actual repair welding, the repair length should be as long as possible.

### 3.4. Effect of Repair Depth

The LRS and TRS distribution of FE models under different repair depths are shown in Figure 14 and Figure 15. Longitudinal tensile stress gradually grew from 339 MPa to 364 MPa, while transverse tensile stress did not exceed the yield strength.

The temperature variation at WCL, WT, and HAZ under different repair depths are shown in Figure 16. It can be seen that the temperature changes corresponding to three positions were basically the same. The temperature at WCL was the highest, followed by the WT and the temperature at HAZ was the lowest, which was consistent with the actual welding situation.

The LRS and TRS distribution at WCL, WT of FE models under different repair depths were obtained in Figure 17 and Figure 18. The residual stress at WCL increased slowly, then dropped rapidly and next grew slowly. However, the residual stress at WT had a slow upward trend first and then fell off. The residual stress at WCL, WT for different repair depth were basically the same.

The membrane and bending stress were obtained based on the linearization of the longitudinal and transverse residual stress at WCL and WT, respectively, as shown in Figure 19. It can be seen from the histograms that no matter at WCL or WT, the LRS maintained a high tensile stress state (approximately 290 MPa). As the deepening of the repair depth, the transverse membrane stress at WCL and WT gradually increased from 194 MPa to 217 MPa and from 191 MPa to 216 MPa, respectively. In addition, transverse membrane stress for repair welding depth D3 (0.75t) became greater by 13.5% than that for repair welding depth D1 (0.25t). However, both the transverse and longitudinal bending stress stayed at a low stress level. Therefore, the optimal repair welding depth was 0.25t based on the analysis of membrane and bending stress.

Next, the LRS and TRS distribution at path 1 is shown in Figure 20. The LRS remained basically remained unchanged with the growth of repair welding depths. The TRS at HAZ of butt plates increased remarkably after repair welding but was basically the same under different repair welding depths.

According to FEA, among 0.25t, 0.5t, and 0.75t, the best repair depth was 0.25t. In the actual repair welding process, the repair depth should be as shallow as possible.

### 3.5. Effect of Repair Width

When the effect of repair welding width on the residual stress distribution was investigated in this study, the longitudinal and transverse stress nephograms are shown in Figure 21 and Figure 22.

According to Figure 21, as the widening of the repair width, the yield stress area gradually widened, and the compressive stress area at both ends in the transverse direction of the butt plates progressively expanded.

Figure 22 shows that with the growth of repair welding width, the high tensile stress area at both ends of repair welding seam both gradually enlarged.

Figure 23 shows the temperature variation at three positions under different repair widths. It can be seen from the line chart that as the widening of repair welding width, since the original weld toe became the repair welding zone, the temperature variation for the repair welding width of 1.5t (W2) and 2t (W3) were the same, which was consistent with the actual welding situation.

The LRS and TRS distribution at WCL, WT under different repair welding widths is shown in Figure 24 and Figure 25, in terms of the overall stress distribution, the repair width of 2t was the most significance, followed by the repair width of 1.5t, and the repair width of 1t was the minimum.

The membrane and bending stress were obtained by the stress linearization decomposition of the transverse and longitudinal residual stress at WCL and WT, as shown in Figure 26. The longitudinal membrane stress was at a high tensile stress level (approximately 290 MPa). With the increasement of the repair width, the transverse membrane stress progressively heightened; transverse membrane stress for repair welding width W3 (2t) were remarkably increase by 40.5%, compared with that for repair welding width W1 (1t). While the longitudinal and transverse bending stress were both small less distinguished.

According to Figure 27a, it can be seen that the repair welding width hardly affected the LRS distribution. From Figure 27b, whether before or after repair welding, the maximum residual stress always appeared at WT of the butt plates, and with the growth of the repair width, the residual stress continuously heightened.

From the simulation results, the repairing effect of 1t was the best among the repairing widths of 1t, 1.5t, and 2t. In the actual repair welding process, the repair width should be narrower.

## 4. Conclusions

In this study, the dimensional effect of repair welding on residual stress redistribution was investigated with the combination of FEA and experimental tests. Some research conclusions are as follows:The residual stress results of FEA were in good agreement with the blind-hole drilling test, which validated the accuracy of the FE simulation;According to the stress linearization in BS7910, transverse membrane stress for repair welding length L1 (15t), depth D3 (0.75t), and width W3 (2t) became greater by 76.6%, 13.5%, and 40.5%, respectively, compared with that for repair welding length L3 (4t) depth D1 (0.25t) and width W1 (1t);The optimal repair welding length, depth, and width in this paper are 15t, 0.25t, and t, respectively, which conforms with the repair welding principle of “SNL (shallow, narrow, and long)” and which provides a significant guiding role for the repair welding of welded structures in actual production.

## Figures and Tables

**Figure 1 materials-15-06399-f001:**
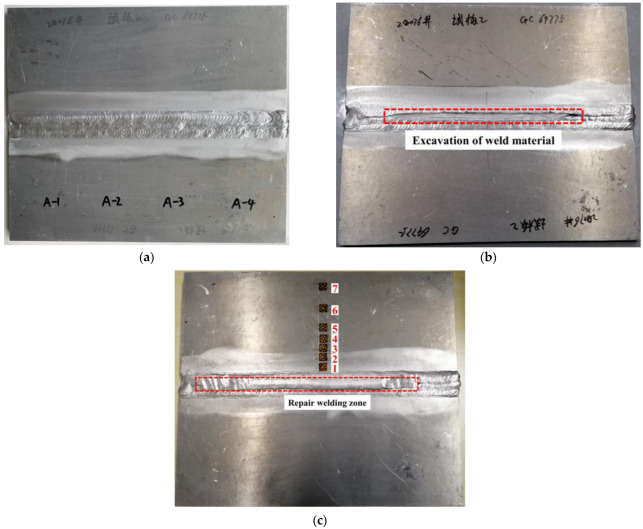
AA6082-T6 aluminum alloy butt plates: (**a**) As-welded; (**b**) Excavation of weld material; (**c**) After repair welding.

**Figure 2 materials-15-06399-f002:**
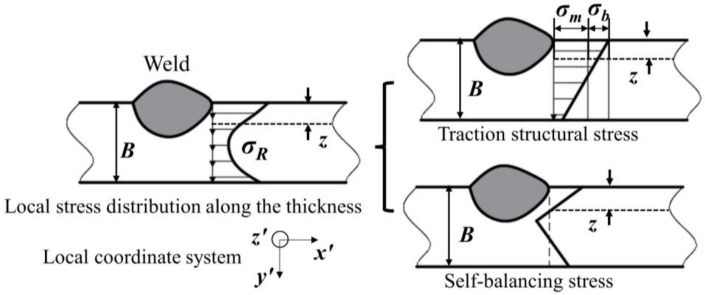
Schematic diagram of stress linearization at the weld toe [39].

**Figure 3 materials-15-06399-f003:**
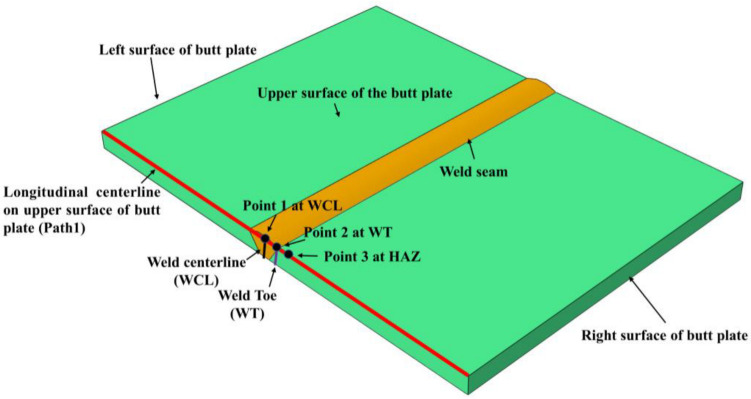
Schematic diagram of residual stress extraction locations.

**Figure 4 materials-15-06399-f004:**
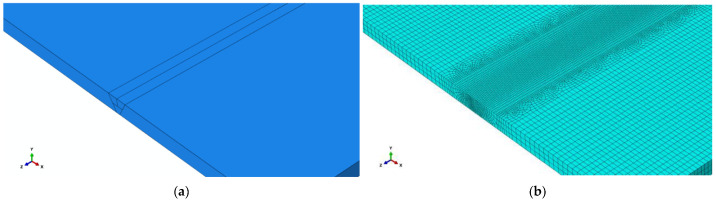
FE models of butt joints: (**a**) 3D modeling; (**b**) Meshing; (**c**) Boundary conditions.

**Figure 5 materials-15-06399-f005:**
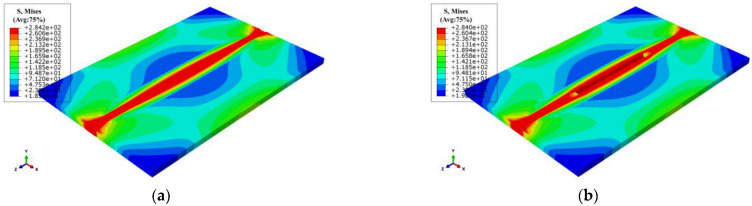
Repair welding simulation: (**a**) The import of welding residual stress; (**b**) Residual stress redistribution.

**Figure 6 materials-15-06399-f006:**
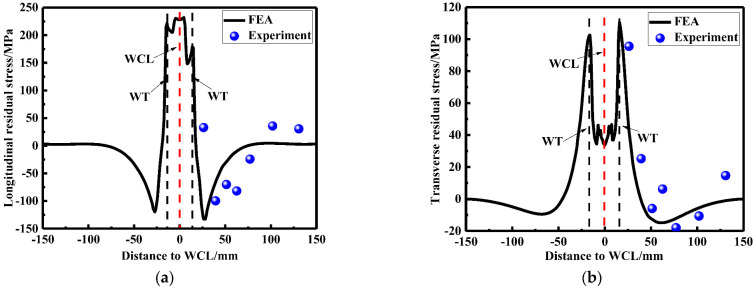
Residual stress comparison obtained by FEA and experimental tests: (**a**) The LRS distribution; (**b**) The TRS distribution.

**Figure 7 materials-15-06399-f007:**
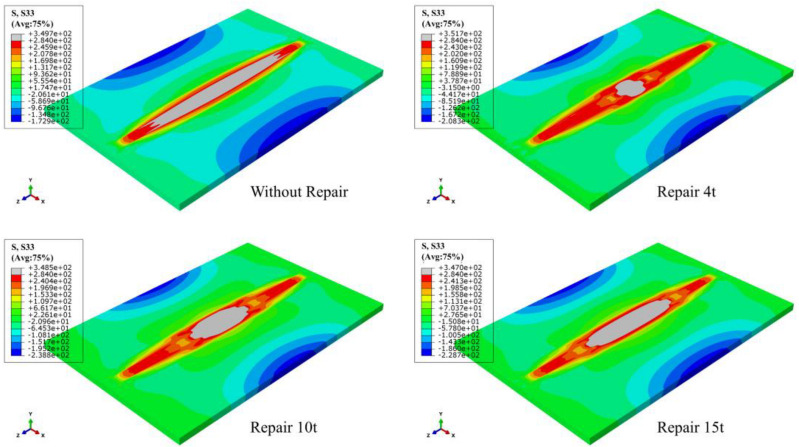
The LRS distribution of FE models under different repair lengths.

**Figure 8 materials-15-06399-f008:**
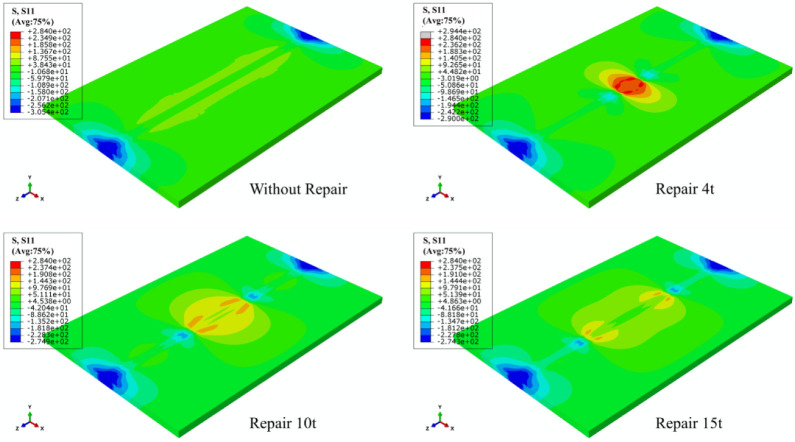
The TRS distribution of FE models under different repair lengths.

**Figure 9 materials-15-06399-f009:**
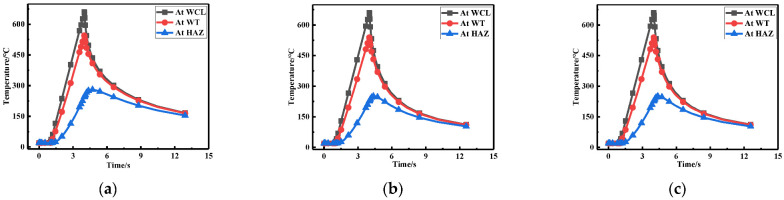
Temperature changes at three positions for: (**a**) Repair length L1; (**b**) Repair length L2; (**c**) Repair length L3.

**Figure 10 materials-15-06399-f010:**
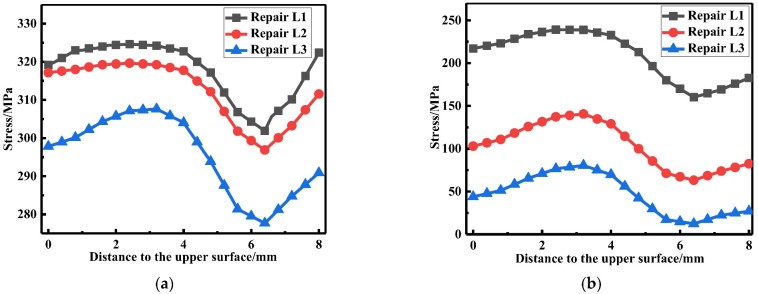
Residual stress distribution at WCL with different lengths: (**a**) The LRS distribution; (**b**) The TRS distribution.

**Figure 11 materials-15-06399-f011:**
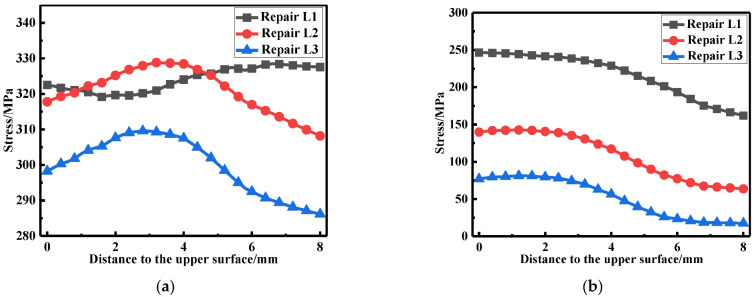
Residual stress distribution at WT with different repair lengths: (**a**) The LRS distribution; (**b**) The TRS distribution.

**Figure 12 materials-15-06399-f012:**
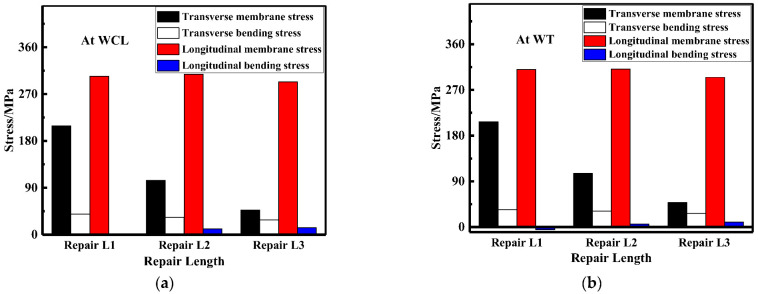
Membrane and bending stress variation for different repair lengths: (**a**) At WCL; (**b**) At WT.

**Figure 13 materials-15-06399-f013:**
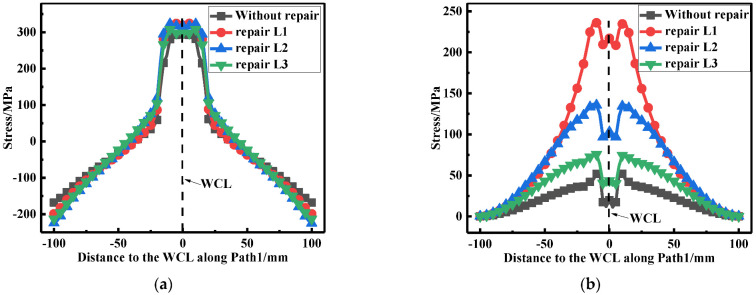
Residual stress distribution at path 1: (**a**) The LRS; (**b**) The TRS.

**Figure 14 materials-15-06399-f014:**
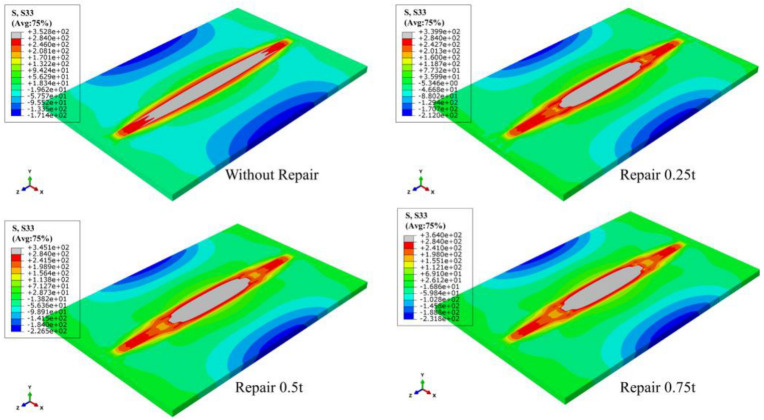
The LRS distribution of FE models with different repair depths.

**Figure 15 materials-15-06399-f015:**
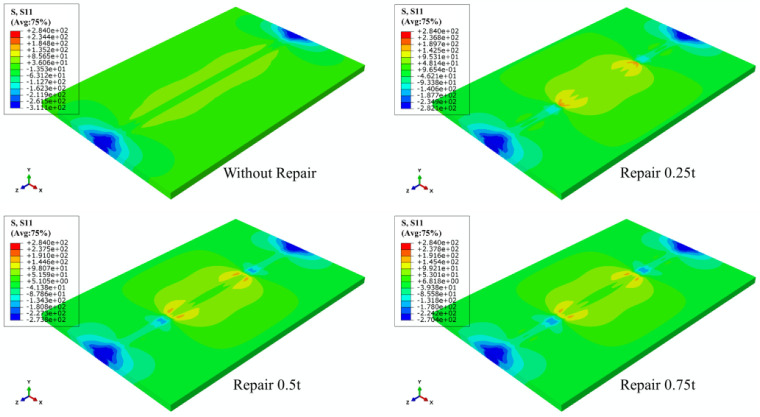
The TRS distribution of FE models with different repair depths.

**Figure 16 materials-15-06399-f016:**
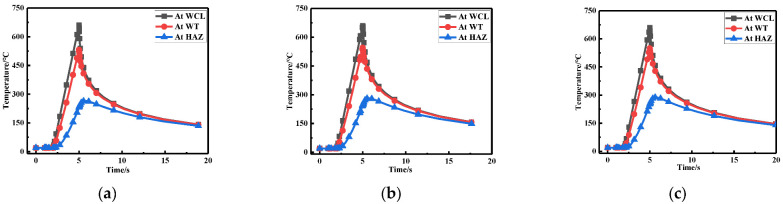
Temperature changes at three positions for: (**a**) Repair length D1; (**b**) Repair length D2; (**c**) Repair length D3.

**Figure 17 materials-15-06399-f017:**
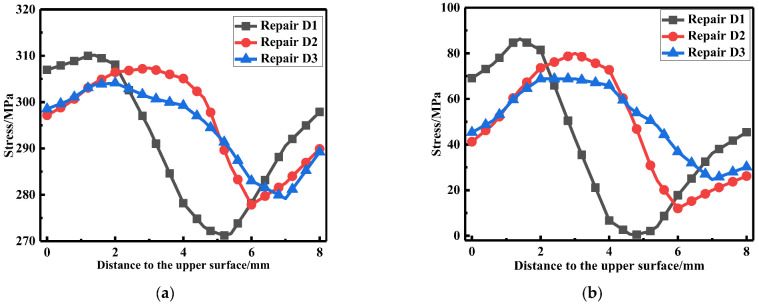
Residual stress distribution at the WCL for different repair depths: (**a**) The LRS distribution; (**b**) The TRS distribution.

**Figure 18 materials-15-06399-f018:**
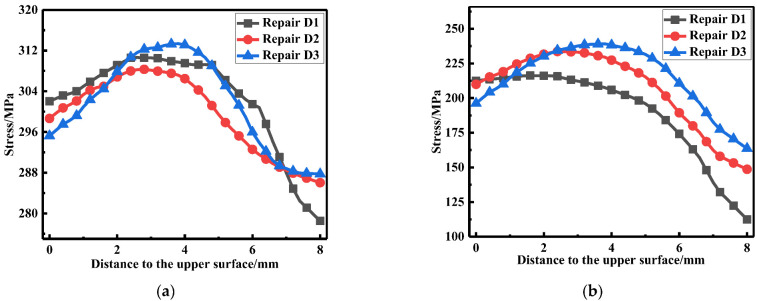
Residual stress distribution at the WT for different repair depths: (**a**) The LRS distribution; (**b**) The TRS distribution.

**Figure 19 materials-15-06399-f019:**
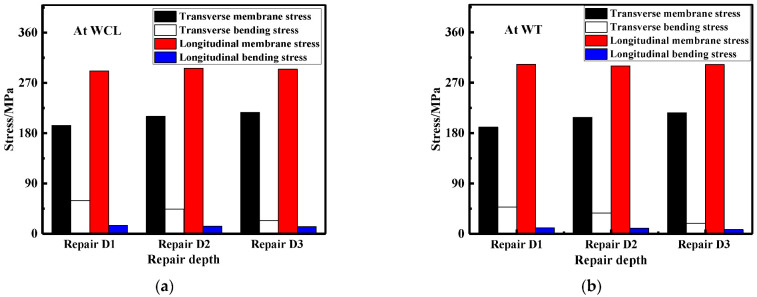
Membrane and bending stress for different repair depths: (**a**) At WCL; (**b**) At WT.

**Figure 20 materials-15-06399-f020:**
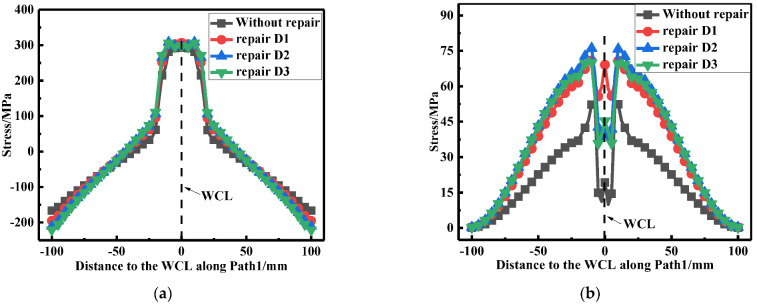
Residual stress distribution at Path 1: (**a**) The LRS distribution; (**b**) The TRS distribution.

**Figure 21 materials-15-06399-f021:**
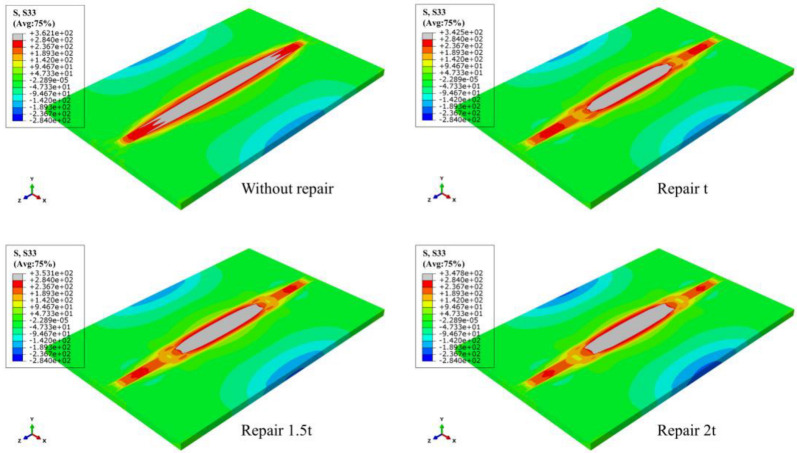
The LRS distribution under different repair widths.

**Figure 22 materials-15-06399-f022:**
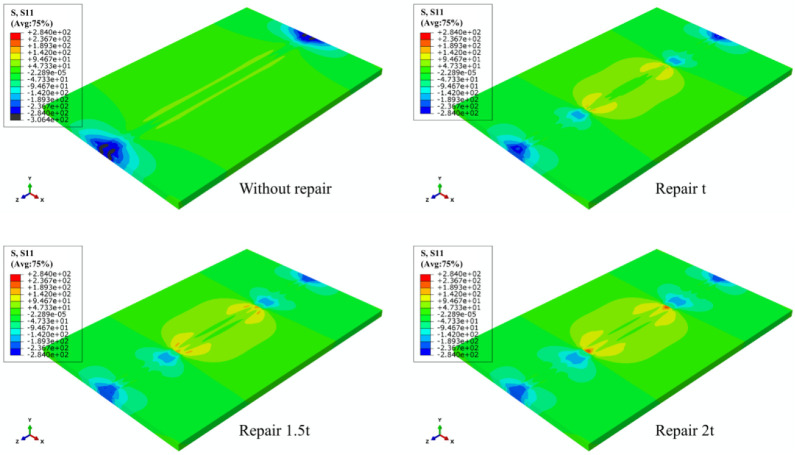
The TRS distribution under different repair widths.

**Figure 23 materials-15-06399-f023:**
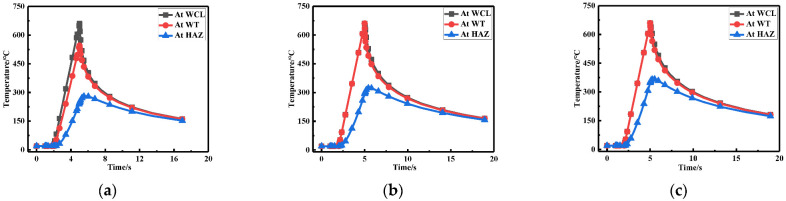
Temperature variation at three positions for (**a**) Repair width W1; (**b**) Repair width W2; (**c**) Repair width W3.

**Figure 24 materials-15-06399-f024:**
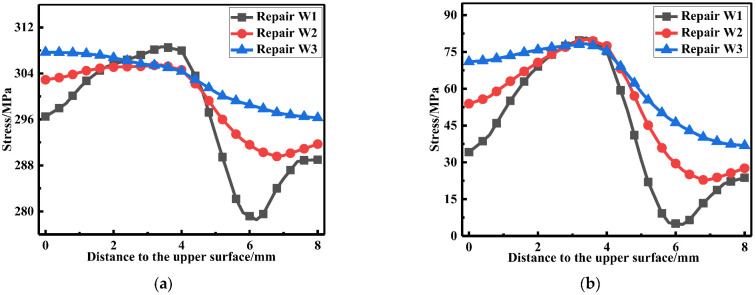
Residual stress distribution at WCL under different repair depths: (**a**) The LRS distribution; (**b**) The TRS distribution.

**Figure 25 materials-15-06399-f025:**
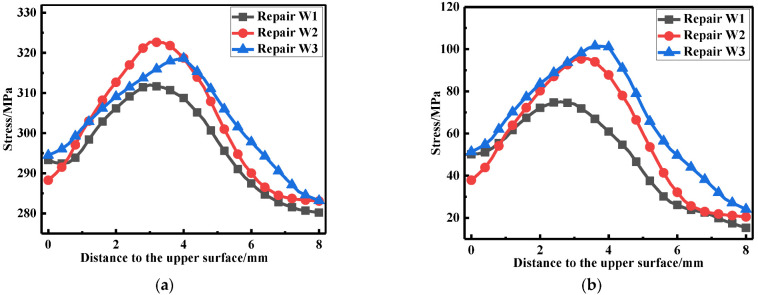
Residual stress distribution on WT with different repair widths: (**a**) The LRS distribution; (**b**) The TRS distribution.

**Figure 26 materials-15-06399-f026:**
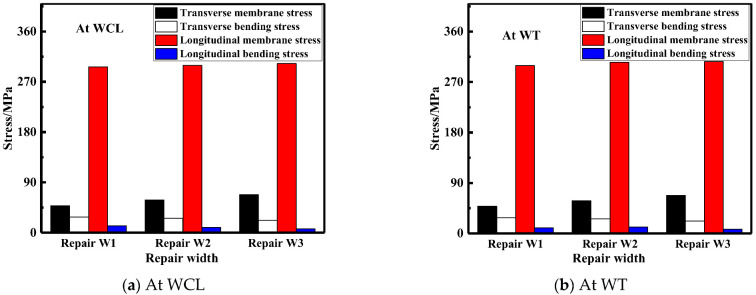
Membrane and bending stress under different repair widths: (**a**) At WCL; (**b**) At WT.

**Figure 27 materials-15-06399-f027:**
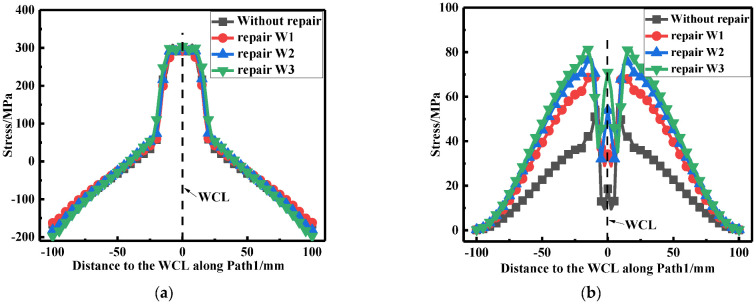
Residual stress distribution at path1 under different repair widths: (**a**) The LRS distribution; (**b**) The TRS distribution.

**Table 1 materials-15-06399-t001:** Chemical compositions of the base metal and weld wire (*wt*.%).

Materials	Si	Fe	Cu	Mn	Mg	Cr	Zn	Ti	Al
AA6082-T6	1.30	0.50	0.10	1.20	1.20	0.15	0.20	0.20	Bal.
ER5356	0.25	-	0.27	0.05	4.50	-	-	-	Bal.

**Table 2 materials-15-06399-t002:** Mechanical properties of AA6082 and ER5356 aluminum alloy.

Properties	Yield Strength *R_el_*/MPa	Tensile Strength *R_m_*/MPa	Elongation *e*/%
AA6082-T6	275	280	9
ER5356	120–190	250–300	15–25

**Table 3 materials-15-06399-t003:** Repair welding length schemes.

Plate Thickness *t*	Repair Width	Repair Depth	Repair Length *L*1	Repair Length *L*2	Repair Length *L*3
8 mm	t	t/2	4t	10t	15t

**Table 4 materials-15-06399-t004:** Repair welding depth schemes.

Plate Thickness *t*	Repair Width	Repair Length	Repair Depth *D*1	Repair Depth *D*2	Repair Depth *D*3
8 mm	t	15t	0.25t	0.5t	0.75t

**Table 5 materials-15-06399-t005:** Repair welding width schemes.

Plate Thickness *t*	Repair Depth	Repair Length	Repair Width *W*1	Repair Width *W*2	Repair Width *W*3
8 mm	t/2	15t	t	1.5t	2t

**Table 6 materials-15-06399-t006:** The specific welding parameters of butt joints.

Melting Temperature *T*_1_/°C	Heating Time *t*_1_/s	Room Temperature *T*_2_/°C
660	3	20
Surface thermal radiation coefficient	Heat transfer coefficient	Weld metal cooling time *t*_2_/s
0.85	0.025	1500

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
