# Peer review of "Residual Stress Redistribution Analysis in the Repair Welding of AA6082-T6 Aluminum Alloy Joints: Experiment and Simulation"

_materials, 2022, doi:10.3390/ma15186399_

Round 1

Reviewer 1 Report

The papers present parametric study of effects of repair weld parameters on resulted residual stresses. There are serious ambiguities that should be addressed:

- More detail about FE simulation should be provided (e.g., heat source modeling, chosen parameters, mesh size, boundary conditions, applied subroutines, etc.)

- More detail about the experiments should be added. At least one figure that shows machined gap before repair welding.

- More detail about FE analyses must be added: number of passes in repair welding for different depths and widths? If all analyses were performed with a single pass, then what we see in results would be more likely due to different in input heats rather than repair weld sizes.

- The English of the manuscript should be chekced and improved.

Reviewer 2 Report

08/17/2021

Review for Journal of Materials

 Manuscript ID: materials-1892118

The authors have presented a sound technical work on  Residual Stress Redistribution Analysis on Repair Welding of 6082-T6 Aluminum Alloy Joints: Experiment and Simulation.

This is a very good and detailed work with a very nicely explained yield phenomenon. However, even with the best of scientific rigour there always remains few queries unanswered. Therefore, I suggest that the thesis shall be ACCEPTED once after MINOR technical revisions and queries as asked below is answered. Some of the queries whose explanation is desired which could enhanced the quality of work is much appreciated and mentioned below.

1.      page 5, line 149 : 3.2. Finite element modelling for repair welding

Meshing

 When meshing the workpiece, how many elements considered? On what basis mesh size is chosen? Have you performed mesh sensitivity? (When meshing the work piece and element the grid size should be adjusted reasonably according to its size, to complete the simulation task within the appropriate calculation time), numerical simulation" do not describe the geometry of the parts, the physical properties assigned to the tool elements, the boundary conditions of the model

2.        Mechanical and metallography test:

The following details should be added in the methodology section: tensile test cross head speed or strain rate, and number of specimens.

3.       Specify how experimental temperature was measure, using thermocouple or Thermal cameras? Provide picture for the same setup.

4.       The literature synopsis provided in the introduction is not wisely balanced: Please include more papers to literature survey

……………………………………………………………………………………………….

Future work: Following points are suggested for future work.

1.       microstructural evolution with an emphasis on phases (composition, fraction, and morphology) will add great value to this research further. It will valuable if the phase analysis is supplemented with X-ray diffraction graphs, It is recommended to analyse the distribution (segregation or homogeneous dispersion) of the alloying elements through energy dispersive spectroscopy method.

2.       Hence the reviewer recommends the usage of statistical tools such as Taguchi, RSM or ANOVA to explore the relationship between the process parameters and properties for future study.

Reviewer 3 Report

The authors studied the residual stress redistribution analysis on repair welding of 6082-t6 aluminum alloy joints: experiment and simulation. The research work is good and is suitable for publication. However, the authors have to do some corrections in the revised manuscript.

1.      The materials should be referred as AA6082-T6 instead of 6082-t6 throughout the manuscript. The changes should be in title also.

2.      The authors should discuss the latest research work reported in literature on repair welding of aluminium alloys.

3.      In conclusion section, the authors reported that “The residual stress results of FEA were in good agreement with blind-hole 337 drilling test, which validated the accuracy of the FE simulation.” How much is the accuracy of FEA simulation?

Round 2

Reviewer 1 Report

The authors have addressed all ambiguities associated with the previous version of the manuscript.